# Integrating Remote Sensing Methods for Monitoring Lake Water Quality: A Comprehensive Review

Anja Batina [ID] and Andrija Krtalić *[ID]

Faculty of Geodesy, University of Zagreb, 10000 Zagreb, Croatia; abatina@geof.hr
* Correspondence: andrija.krtalic@geof.unizg.hr

**Abstract:** Remote sensing methods have the potential to improve lake water quality monitoring and decision-making in water management. This review discusses the use of remote sensing methods for monitoring and assessing water quality in lakes. It explains the principles of remote sensing and the different methods used for retrieving water quality parameters in complex waterbodies. The review highlights the importance of considering the variability of optically active parameters and the need for comprehensive studies that encompass different seasons and time frames. The paper addresses the specific physical and biological parameters that can be effectively estimated using remote sensing, such as chlorophyll-$\alpha$, turbidity, water transparency (Secchi disk depth), electrical conductivity, surface salinity, and water temperature. It further provides a comprehensive summary of the bands, band combinations, and band equations commonly used for remote sensing of these parameters per satellite sensor. It also discusses the limitations of remote sensing methods and the challenges associated with satellite systems. The review recommends integrating remote sensing methods using in situ measurements and computer modelling to improve the understanding of water quality. It suggests future research directions, including the importance of optimizing grid selection and time frame for in situ measurements by combining hydrodynamic models with remote sensing retrieval methods, considering variability in water quality parameters when analysing satellite imagery, the development of advanced technologies, and the integration of machine learning algorithms for effective water quality problem-solving. The review concludes with a proposed workflow for monitoring and assessing water quality parameters in lakes using remote sensing methods.

**Keywords:** water quality monitoring; decision-making; optically active parameters; computer modelling; band combinations; sensors

## 1. Introduction

Lakes are important ecosystems that sustain a wide range of species and are essential for a variety of industries and human activities [1,2]. However, eutrophication, human exploitation, and climate change all have a negative impact on lake water quality and the general health of lake ecosystems [3–7]. To address this, remote sensing evolved as an effective method for monitoring and analysing worldwide water quality. This method, which collects spectrum data from aerial and satellite platforms, has been used since the 1970s to assess the physical, chemical, and biological characteristics of water quality.

Traditional methods of monitoring water quality through in situ measurements and laboratory analysis are time-consuming and costly, with limited geographical and temporal variability [8]. Remote sensing is a cost-effective and time-saving method that provides unique spatial information and data continuity for large-scale areas and inland water-bodies [9–12]. It can be combined with conventional methods to address the constraints of in situ methods [13]. Remote sensing methods and databases are highly useful for gathering data on lake ecological indicators, particularly in unstudied lakes with minimal in situ monitoring. Remote sensing, with sufficient in situ validation, may offer near-real-time information on lake changes, such as algae blooms or droughts. Interdisciplinary

collaboration and validation may improve the accuracy and efficiency of remote sensing for waterbody evaluation and management [11], while requiring less time, effort, and money [14,15].

The conventional method for assessing water quality includes three types of parameters: (1) physical parameters such as water temperature (WT), transparency (Secchi disk depth (SDD)), salinity, turbidity, total suspended matter (TSM), coloured dissolved organic matters (CDOM), odour, and electrical conductivity (EC); (2) chemical parameters such as pH, dissolved oxygen (DO), chemical oxygen demand (COD), biochemical oxygen demand (BOD), total organic carbon (TOC), dissolved organic carbon (DOC), total nitrogen (TN), ammonia nitrogen ($NH_3$-N), nitrate nitrogen ($NO_3$-N), total phosphorus (TP), orthophosphate ($PO_4$), heavy metal ions, and nonmetallic toxins; and (3) biological parameters such as chlorophyll-$\alpha$ (chl-$\alpha$), total bacteria, and total coliforms. The analysed water quality parameters are divided into two groups using remote sensing methods. The initial categorization consists of parameters with active optical characteristics, including chl-$\alpha$, TSM, and CDOM. These parameters affect the radioactive transfer process of waves by modifying the absorption of the spectrum. The second categorization includes parameters without defined optical properties, such as TN, TP, and DO. These parameters are commonly examined using statistical correlations with optically active parameters [15]. The review covers six optically active water parameters, including chl-$\alpha$, turbidity, SDD, WT, salinity, and EC.

This paper presents a comprehensive review of the current research status and developments in the use of remote sensing methods to monitor lake water quality. The review examines numerous research projects that evaluate water quality using remotely sensed data, emphasizing the potential use of these results for environmental researchers. It aims to offer a centralized resource for academics to obtain insights into present practices and suggest areas for improvement or future contributions. The review focuses on optically active physical and biological parameters that may be retrieved using satellite imagery, and it includes data from specialists in lake hydrology, biology, ecology, and chemistry. It also addresses the dependability of data representation by shifting from point to raster representation. The objectives of this paper are to (1) provide a bibliometric analysis, (2) provide insight into the current state of remote sensing methods for monitoring water quality in lakes, (3) summarize methods for retrieving water quality parameters based on remote sensing used in the literature, (4) provide a comprehensive summary of the bands, band combinations, and band equations commonly used for remote sensing of water quality parameters per satellite sensor, (5) address the importance of optimizing grid selection and time frame for in situ measurements, and (6) propose a workflow for monitoring and assessing water quality parameters in lakes using remote sensing methods. In addition, this review discusses the elements that influence the correlation between water quality parameters and satellite imagery, as well as possible solutions and limits to the challenges of remote sensing water quality assessment in lakes. Overall, this review adds new knowledge to the field and encourages further research and innovation in remote sensing methods for water quality monitoring.

## 2. Bibliometric Analysis

Long ago, remote sensing was acknowledged as a method for global tracking of inland water quality. Airborne and satellite spectral data collection has been used since the early 1970s to analyse a broad collection of water quality parameters [16] (Figure 1). Previous reviews [14,15,17–19] have offered excellent summaries of hundreds of publications presenting models for evaluating the biological, chemical, and physical properties of complex waterbodies published by scientists during the past 50 years. General trends in this sense were revealed after extensive bibliometric research of the Elsevier Scopus database (conducted in May 2024). Database titles, keywords, and abstracts from 1977 until 2023 for the terms 'remote sensing', 'water quality', and 'lake' in the English language were searched. The search found 29,901 unique publications published for terms 'water quality' and 'lake' and 1788 unique publications published for terms 'remote sensing', 'water quality', and

'lake'. Globally, the number of publications employing remote sensing for lake water quality falls significantly behind those that do not, as shown in Figure 1.

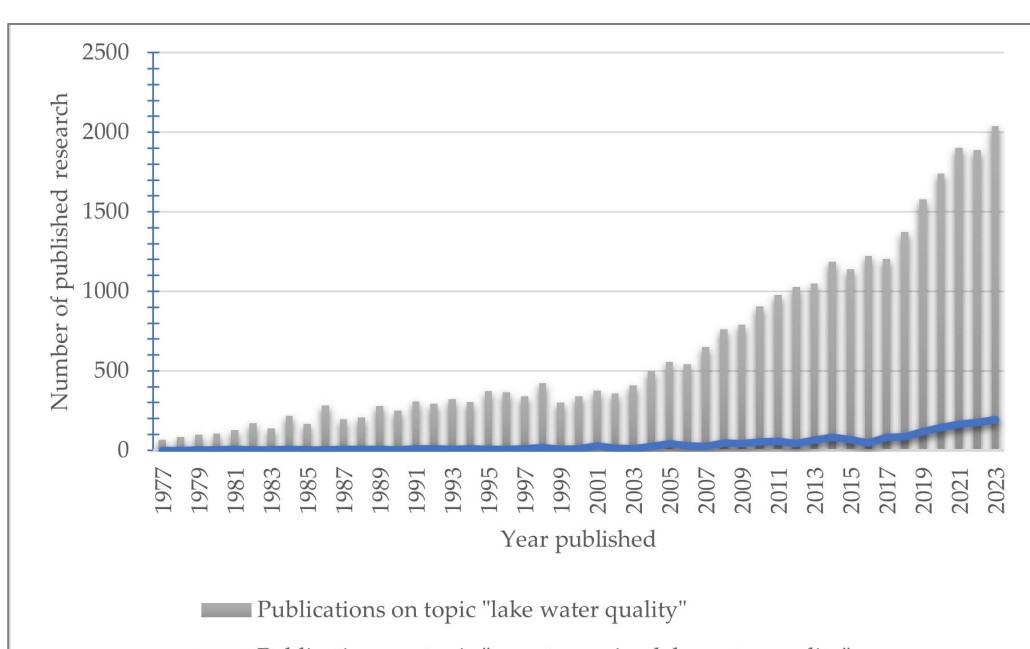

**Figure 1.** Number of publications retrieved from Elsevier Scopus on the search topic "(remote sensing) lake water quality".

Since the 1970s, papers describing the monitoring and assessment of water quality in lakes utilizing remote sensing methods have arisen (Figure 1). Publication numbers for the specific dataset follow a power law distribution, with a similar increase after 2008. The greatest increase in remote sensing studies from one year to the next happened after 2008, coinciding with the availability of free Landsat imagery. This conclusion is consistent with prior research indicating that the publication of the Landsat archive led to an increase in the frequency and scope of EO studies in different domains [20] and has resulted in a more comprehensive understanding of inland waterbodies to concentrate on demanding scientific problems and expanding research scales. Research on lakes based on remote sensing has seen a significant increase in publications in the past decade (2014–2023) compared to the previous 37 years (1977–2013). The majority of the publications in the bibliometric analysis were published in the United States and China, with the rest originating from various countries across Europe and from India, Canada, Japan, and Australia, as seen in Figure 2.

The literature review suggests that the recent advancement in evaluating inland water quality through remote sensing initiatives can be attributed to the challenges associated with remote sensing of complex waterbodies and the limited availability of suitable sensors for this purpose (viz., hyperspectral airborne or space-borne remote sensing that captures extensive spatial and spectral information, especially in small lakes) [21]. This review uses the 'System A' lake typology of the WFD [22,23], which categorizes lakes based on four abiotic characteristics. The focus of the review is on categorizing lakes based on mean depth and surface area (Table 1). The complex bio-optical properties of a static waterbody (vegetation and pollutants) create problems in establishing an internal correlation between spectral responses and optically inactive water quality parameters. In tandem with the increase in remote sensing data accessibility during the past decade, in situ data accessible for model calibration and validation has increased. Modern databases offered by government agencies, nongovernmental organizations, and scholars provide a variety of freely accessible in situ data. In Europe, these include Eye on Water (www.eyeonwater.org

(accessed on 16 October 2023)) and Seen-monitoring (www.seen-transparent.de (accessed on 16 October 2023)) [18]. These established databases can be enhanced with innovative datasets collected through citizen science activities. In this way, data continuity is offered, resulting in cost and time savings for researchers, and a multitude of samples for calibration and validation of derived models.

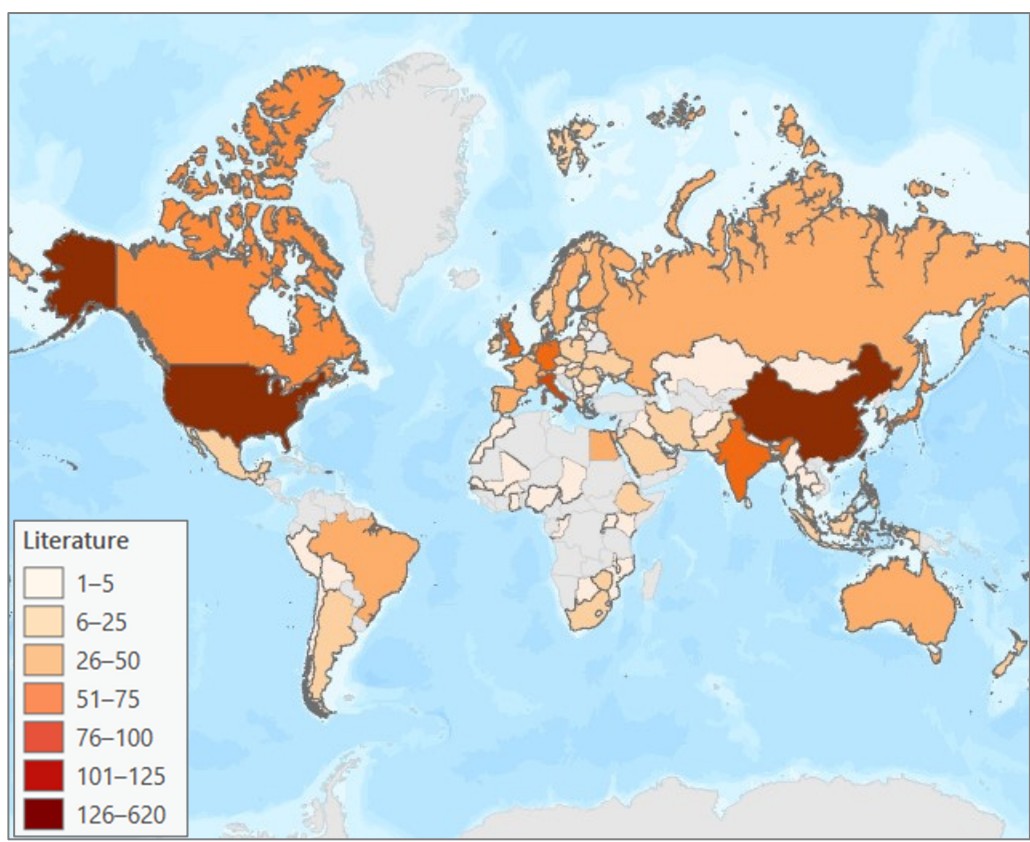

**Figure 2.** A map showing the literature count per country of origin based on the bibliometric analysis.

**Table 1.** WFD lake typology [22,23].

| Lake Parameter | Value | Description |
|---|---|---|
| Depth | <3 m | Very shallow |
| | 3–15 m | Shallow |
| | >15 m | Deep |
| Surface area | <1 km$^2$ | Small |
| | 1–10 km$^2$ | Medium |
| | 10–100 km$^2$ | Large |
| | >100 km$^2$ | Very large |

## 3. Materials and Methods

The principle of using remote sensing methods to assess water quality involves creating models based on spectral responses and in situ measurements of water quality parameters. Locations of in situ measurements should consider hydrodynamic models and serve as a basis for comprehensive lake-wide analysis via remote sensing by choosing bands, band combinations, and band equations for retrieving optically active parameters. These models are calibrated and validated using in situ measurements and used for comprehensive analysis of water quality over a larger area and longer period [15]. Bio-optical methods use the correlation between a waterbody's optical properties and its optically active parameters to

assess water quality. The performance of water quality retrieval models depends on spatial factors and the inherent optical properties of the region [17]. Optically active parameters interact with light and can be obtained through direct retrieval [24], while optically inactive parameters can be inferred from measurable water quality parameters [25].

There are five methods commonly used for monitoring and assessing water quality using remote sensing imagery: empirical, semi-empirical, semi-analytical, analytical, and machine learning (ML). Each method has its own characteristics and complexity.

### 3.1. Analytical Methods

Analytical models, also known as physical models, are used to determine spectral reflectance by analysing the optical properties of water and the atmosphere. These models are based on physics [18] and use inherent and apparent optical properties to model surface water reflectance and determine the concentration of constituents [14]. They link water quality parameters with water-leaving radiance using radiation transmission theory [26]. The analytical method can identify all water parameters simultaneously, but it requires accurate measuring instruments and has high application costs [27]. Model development is challenging due to differences in spectral resolutions between satellite sensors and ground measurements. The analytical method is infrequently used for all water quality parameters [15] and requires theoretical breakthroughs to create more generalized models; nevertheless, it has good portability [27]. Studying the complex optical characteristics of water quality parameters can improve the accuracy of analytical methods [15].

### 3.2. Semi-Analytical Methods

Analytical and semi-analytical models are used to study the physics-based optical properties of water and the atmosphere [28]. Semi-analytical methods simplify analytical models and require statistical analysis [29]. Theoretical values are calculated by modelling the optical properties of a waterbody [28]. Some models are based on water column radiative transfer and use inversion and look-up tables to match spectral signatures and predict water quality parameters [30]. Semi-analytical models using in situ observations are common for remotely sensing inland water quality [17]. However, model development is challenging and requires knowledge of atmospheric correction and substantial in situ sampling [18]. These models have been successfully applied on broad spatiotemporal scales to retrieve optically active parameters, such as chl-$\alpha$ and SDD [15].

### 3.3. Empirical Methods

The empirical method is a statistical approach that uses regression analysis to establish relationships between water quality parameters and spectral response values [14,15]. This method is used to derive distinctive bands or band combinations and create a water quality inversion model [31]. Empirical methods include linear regression, band combination, and principal component analysis. However, these methods lack physical mechanisms and multitemporal validity, resulting in uncertainty and limited applicability. Inland waterbodies, which are optically complex, often require multivariate regression [32]. Empirical models also rely on in situ data and may be affected by changes in downwelling irradiance and water surface conditions [33]. Despite these drawbacks, the empirical method is preferred for its simplicity, low computational needs, and ability to account for specific waterbody properties [17,32]. It is commonly used to assess turbidity, chl-$\alpha$, and trophic status [16].

### 3.4. Semi-Empirical Methods

Semi-empirical methods combine empirical and analytical methods to correlate water quality parameters with remote sensing data [27]. These methods involve statistical and measured spectral analysis to select characteristic bands and develop models [34]. They use physical and spectral data to create algorithms that correlate with measured parameters. However, their validity is limited to a specific range of optical water quality data [33]. Semi-empirical models do not model the inherent optical properties of waterbodies like

semi-analytical models, but they improve the spectral properties of parameters and reduce noise. Physically based semi-empirical models are more generalizable but require sensors with properly positioned band centres and sufficient spectral resolution [18]. Spectral band ratios and shape algorithms are commonly used due to their generalizability and ease of implementation, although they assume consistent water and atmospheric conditions. Spectral band ratios and spectral shape methods are better for assessing regional water quality parameter distributions than precise estimates [32]. The temporal and spatial applicability of semi-empirical methods is limited by the availability of in situ measured data. These methods are often used to assess parameters such as chl-$\alpha$, SDD, and turbidity [35–37].

*3.5. Machine Learning (ML) Methods*

The use of ML methods in remote sensing has been increasing [38,39]. ML methods, such as partial least squares regression (PLSR), support vector regression (SVR), artificial neural networks (ANN), deep neural networks (DNN), and convolutional neural networks (CNN), have shown promise in accurately estimating water quality parameters in remote sensing. Traditional ML algorithms like PLSR and SVR [40,41], and deep learning (DL)-based methods like ANN, DNN, and CNN [18] excel at solving complex nonlinear problems. ANN models require large training samples, while SVM models are suitable for small samples and nonlinearity [40]. CNN models are particularly effective for classifying hyperspectral images [42].

ML models are limited by the data used to train them and require distinct training and testing datasets. In order to ensure that representative samples of a data set are selected, a random split of 70% training data and 30% testing data should be used, according to [32]. They can capture complex and nonlinear relationships between water quality parameters and remotely sensed reflectance when given appropriate inputs [18]. However, ML methods have downsides, such as the need for a lot of training data, the challenge of combining features from different spectral, spatial, and temporal information, and the potential for unexplained solutions or ill-posed problems [43]. Despite these challenges, the use of ML in remote sensing for water quality estimation has become more popular due to algorithm development, sensor systems, computing power, and data accessibility [44]. DL methods have been found to outperform other remote sensing methods in estimating water quality parameters such as chl-$\alpha$, turbidity, $NO_3$-N, and $PO_4$-P [32,45].

**4. Optically Active Water Quality Parameters**

Water molecules have properties such as scattering, reflecting, and absorbing the electromagnetic spectrum (Figure 3), which can create challenges for remote sensing in aquatic environments [46]. These properties limit optical remote sensing to the visible area of the electromagnetic spectrum [46], although the near infrared region can provide some information [13], especially in shallow water. Optically active parameters, such as chl-$\alpha$, interact with light and modify radiation in the water column through absorption and scattering processes [25]. Remote sensing can accurately measure these parameters and other water quality parameters without the errors associated with in situ measurements. In optically shallow water, the reflected light also contains information about the bottom substrate and bathymetry [19]. The apparent optical properties depend on water quality and radiation geometry [47], and retrieval models can be built based on the interaction between inherent optical properties and remote sensing reflection. Remote sensing has been successful in measuring various optically active water quality parameters, including chl-$\alpha$, SDD, turbidity, salinity, and WT [24,32,45,48–50], but there are challenges in estimating parameters with weak optical properties, such as pH, DO, nutrients, and heavy metals [31,32,50–52]. However, it is possible to estimate these parameters by establishing correlations with optically active parameters. The review emphasizes the importance of considering the variability of optically active biological (chl-$\alpha$) and physical parameters (SDD, EC, turbidity, salinity, and WT), the correlation with satellite imagery, and the need

for comprehensive studies that encompass different seasons and time frames to ensure accurate assessments and effective management of water resources.

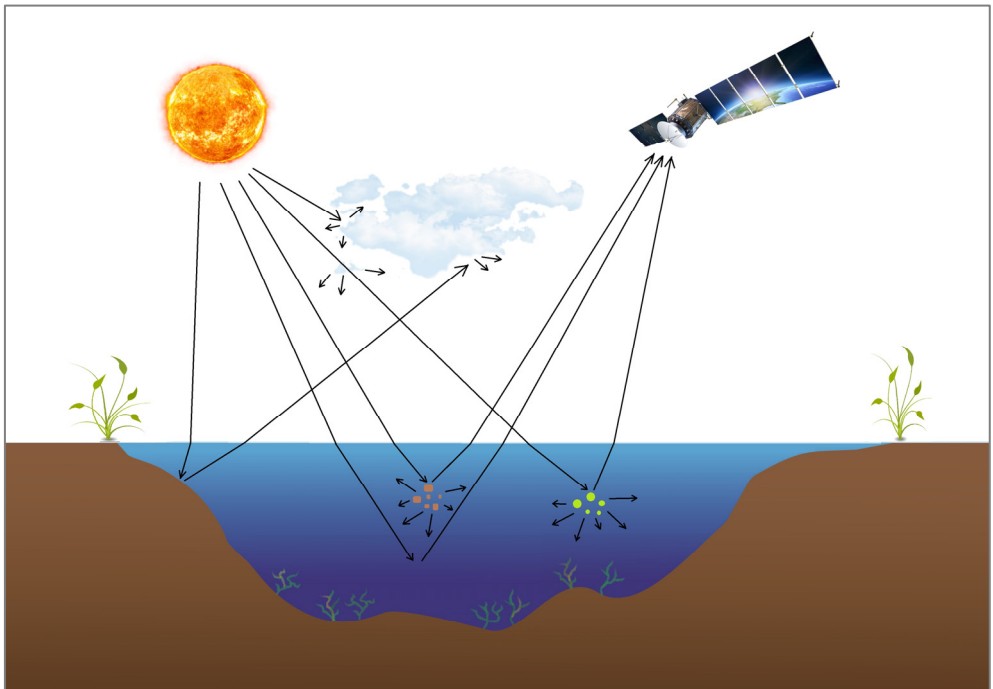

**Figure 3.** Schematic overview of the path of the electromagnetic spectrum from the sun to a waterbody and a sensor.

*4.1. Chlorophyll-α (Chl-α)*

Photosynthesis is a vital process for plants and other photosynthetic organisms, as it allows them to convert light energy into usable energy. Chlorophyll, specifically chl-α, is the most common pigment involved in photosynthesis [17]. It plays a crucial role in water-body primary productivity, trophic status, and nutrient levels. However, excessive chl-α concentrations can lead to harmful algal blooms, particularly those caused by phycocyanin-producing cyanobacteria, which can be toxic to humans and wildlife [53]. The increase in harmful algal blooms worldwide is attributed to anthropogenic nutrient loading and climate change [54]. Therefore, it is important for local authorities to monitor and forecast these blooms.

Remote sensing methods, such as satellite and aerial imagery, can be used to assess chl-α concentrations in waterbodies. Due to sunlight-induced fluorescence, the chl-α spectrum peaks at 680 nm in oligotrophic to mesotrophic aquatic environments [55]. Narrow bands of imagery are needed for remote sensing chl-α concentration and its geographical and temporal fluctuations [56]. Table S1 lists selected remotely taken measurements of chl-α using various sensors and spectral bands, band ratios, and band combinations. As seen from Table S1, a conducted literature review showed Landsat-5 TM and Envisat MERIS as most suitable and popular for chl-α evaluation due to their easy accessibility, temporal coverage, and spatial resolution, making them a good choice.

Based on the summary given in Tables S1 and S2, various methods, including analytical [57–60], semi-analytical [35,61–63], empirical [10,31,51,64–67], semi-empirical [68–70], ML [41,71], and NNs [24,67,72], have been employed to analyse remote sensing data and estimate chl-α concentrations in lakes. These methods utilize different approaches, such as measuring the optical properties of water, combining field and remote sensing data, establishing statistical relationships, and modelling complex relationships. The findings of Wu et al., 2009 [72] and Song et al., 2011 [67] demonstrated that NN models outperformed empirical models in utilizing spectral information and model reliance and highlighted the superior accuracy of utilizing NN models in water quality monitoring and manage-

ment efforts compared to empirical regressions. The application of these methods has provided valuable insights into the dynamics and distribution of chl-$\alpha$ in different aquatic ecosystems, contributing to a better understanding of algal abundance and informing management strategies.

According to the outline provided in Table S2, waterbodies with a narrow range of measured chl-$\alpha$ values (chl-$\alpha$ concentrations between 0.01 and 11 µg/L) show a notable correlation with specific imagery from satellite and airborne sensors. Landsat-5 TM imagery revealed moderate correlations ($R^2$ = 0.513, 0.53, and 0.72) for a lake in Turkey [73] with in situ values ranging from 0.62 to 3.99 µg/L, a reservoir in Arkansas, USA [74], with in situ values ranging from 1.4 to 10 µg/L, and a lake in Italy [75] with in situ values ranging from 1.11 to 4.57 µg/L, respectively. There is a moderate correlation between MIVIS aerial hyperspectral imagery ($R^2$ = 0.71) for a lake in Italy [58] with in situ values ranging from 0.75 to 4.3 µg/L, and between EO Hyperion-1 satellite hyperspectral imagery ($R^2$ = 0.705) for a lake in Guatemala [68] with in situ values ranging from 1.01 to 10.91 µg/L. There is a strong correlation between a reservoir in Arkansas, USA [76] when compared using Landsat-5 TM ($R^2$ = 0.84) and in situ values ranging from 1 to 7 µg/L in four months throughout the year. A lake in Germany [70] correlates strongly with CASI and HyMap aerial hyperspectral imagery ($R^2$ = 0.89) with in situ values ranging from 1 to 3 µg/L collected from May to September.

Imagery from satellite and airborne sensors has shown a moderate to strong correlation with waterbodies with chl-$\alpha$ concentrations in the medium range of 0.07 and 40 µg/L. A moderate correlation has been discovered between chl-$\alpha$ and EO Hyperion-1 hyperspectral imagery ($R^2$ = 0.59) in a lake in Italy [57], with in situ values from 0.5 to 12 µg/L measured in one month. Terra MODIS imagery has a moderate correlation ($R^2$ = 0.632) for a lake in China [72] with chl-$\alpha$ in situ values from 5.2 to 33.9 µg/L during a 4-month study, and Landsat-5 TM imagery ($R^2$ = 0.72) for a lake in Italy [75] with in situ values from 4.63 to 11.35 µg/L measured in one month. Strong correlations were observed between Landsat-5 TM and lakes in Spain [77], China [67], and Italy [78], with $R^2$ values of 0.82, 0.98, and 0.999, respectively. However, only four samples were used for the research of the lake in Italy [78], and in situ values ranged from 5.5 to 7.7 µg/L, while in situ values in the lake in Spain ranged from 0.4 to 20 µg/L in a 6-year study and in situ values in the lake in China ranged from 5 to 30 µg/L in one month.

A correlation has been found between imagery from satellite and airborne sensors and chl-$\alpha$ concentrations in waterbodies, which vary considerably in the wide range from 0.01 to 250 µg/L. A moderate correlation was observed between chl-$\alpha$ and Landsat-5 TM with a coefficient of correlation $R^2$ of 0.705 across 42 lakes in Michigan, USA [79]. The chl-$\alpha$ in situ values in Michigan lakes [79] varied from 0.2 to 87 µg/L over a period of six months. There is a strong correlation ($R^2 \geq 0.8$) between chl-$\alpha$ and several satellite and airborne imagery sources, such as Ikonos OSA, AISA, CASI, HyMap, PROBA-CHRIS, Landsat-7 ETM+, Envisat MERIS, Sentinel-2 MSI, and Sentinel-3 OLCI. This correlation was seen in 15 studies from Table S2 conducted across study periods ranging from one month to 13 years, with maximum chl-$\alpha$ levels reaching 120 µg/L.

Waterbodies with a very wide range of chl-$\alpha$ concentrations between 0.01 and 700 µg/L show a strong correlation with specific satellite images. A strong correlation ($R^2$ = 0.85) was discovered in a 3-year study, including 13 reservoirs in Oklahoma, USA [80]. The research used PlanetScope, Sentinel-2 MSI, and Landsat-8 OLI data in conjunction with chl-$\alpha$ in situ measurements ranging from 0.6 to 540 µg/L, where Sentinel-2 MSI showed the highest correlation with chl-$\alpha$ in situ values. A 6-year study conducted on nine waterbodies in the United States, Australia, and China [41] revealed a strong correlation ($R^2$ = 0.91) between Sentinel-3 OLCI images and in situ readings ranging from 2.8 to 285.5 µg/L. A 1-month study conducted on 15 lakes in Minnesota, USA [81], revealed a strong correlation ($R^2$ = 0.99) between chl-$\alpha$ concentrations ranging from 1.8 to 397 µg/L and Terra MODIS images.

The literature summarized in Table S2 provides information on the most commonly used sensors for assessing chl-$\alpha$ from satellite and airborne multispectral and hyperspectral imagery. The Landsat-5 TM sensor is frequently utilized and has an average $R^2$ value of 0.76 based on eight studies [67,73–79]. However, several other sensors have achieved better results for assessing chl-$\alpha$. Satellite multispectral sensors such as Sentinel-3 OLCI and Ikonos OSA, satellite hyperspectral sensors such as Envisat MERIS and PROBA-CHRIS, and airborne hyperspectral sensors such as CASI, HyMap, and AISA have achieved the best $R^2$ values (>0.88) for assessing chl-$\alpha$. Very shallow waterbodies (<3 m) have the highest $R^2$ values of 0.84 for retrieving chl-$\alpha$ from satellite and airborne imagery. The best results, with an $R^2$ value of 0.93, are achieved for medium-sized waterbodies (1–10 km$^2$). Additionally, studies lasting longer than one year have achieved better results, with an $R^2$ value of 0.85. The most successful methods for retrieving chl-$\alpha$ from satellite and airborne imagery are empirical, NN, ML, and nonlinear regression, with $R^2$ values exceeding 0.93.

The most effective sensors for retrieving chl-$\alpha$ in small waterbodies (<1 km$^2$) are Landsat-5 TM, airborne hyperspectral CASI, and HyMap. For medium waterbodies (1–10 km$^2$), the most effective satellite sensor is hyperspectral Envisat MERIS. Ikonos OSA is the recommended multispectral sensor for large waterbodies (10–100 km$^2$), while multispectral Sentinel-3 OLCI and hyperspectral Envisat MERIS (satellite platforms) are most effective for very large waterbodies (>100 km$^2$), according to the literature in Table S2.

The most effective sensors for retrieving chl-$\alpha$ in very shallow waterbodies (<3 m) are multispectral Sentinel-3 OLCI, satellite hyperspectral Envisat MERIS, and airborne hyperspectral CASI and HyMap. In shallow waterbodies (3–15 m), the recommended satellite sensor is the hyperspectral Envisat MERIS. The most effective sensors for retrieving chl-$\alpha$ in deep waterbodies (>15 m) are multispectral Ikonos OSA, airborne hyperspectral CASI, and HyMap.

*4.2. Turbidity*

Turbidity is a measurement of the amount of suspended and dissolved particles in water that cause light to scatter [19]. High turbidity levels can reduce water transparency and carry contaminants and nutrients, impacting primary production, aquatic plant growth, and water quality in lakes [82]. Remote sensing is used to map turbidity concentrations and their variations over time and space. Various methods, including the empirical method [83–86], have been used to measure turbidity in lakes and reservoirs. Studies have shown that ML methods, such as NNs, can provide accurate predictions of turbidity concentrations [67,85]. Table S3 lists selected remotely taken measurements of turbidity using various sensors and spectral bands, band ratios, and band combinations. Integrating ML into turbidity concentration studies has the potential to enhance understanding of water quality dynamics in aquatic systems.

Waterbodies with turbidity levels in a narrow range from 0.1 to 20 NTU have a notable correlation with certain satellite images. Landsat-5 TM images showed a moderate correlation ($R^2$ = 0.537) for a lake in Tennessee, USA [79]. The in situ turbidity values in a lake in Tennessee, USA [79], vary from 4.1 to 20 NTU in one month.

Waterbodies with turbidity levels in the medium range of 0.1 to 100 NTU have a significant correlation with certain satellite images. There is a strong correlation between turbidity and Landsat-5 TM ($R^2$ = 0.822) in a lake in Turkey [73], with in situ values varying from 2.9 to 33.5 NTU during a 1-month period. There is a strong correlation between turbidity and PROBA-CHRIS ($R^2$ = 0.9) as well as turbidity and Landsat-5 TM/Landsat-7 ETM+ ($R^2$ = 0.85) in a reservoir in Cyprus [87]. In situ turbidity levels ranged from 7.94 to 26.3 NTU during a period of six months (April–October 2010). The strongest correlation was identified between turbidity and Terra ASTER data for a lake in Egypt [83], with an R-squared value of 0.998, over the in situ range of 0–85 NTU measured during a 2-month period.

Waterbodies with turbidity levels in the wide range of 0.1 to 200 NTU exhibit a notable correlation with certain satellite images. Analysed Landsat-8 OLI data indicated a moderate

correlation ($R^2 = 0.642$) for turbidity levels ranging from 13.5 to 117 NTU in a reservoir in Columbia [88] during a 1-month period. A strong correlation has been found between turbidity and Landsat-5 TM images for a lake in China [67] and a reservoir in China [89] ($R^2 = 0.98$ and 0.937, respectively). Research on a lake in China [67] spanned one month and measured in situ values between 5 and 180 NTU. Research conducted on a reservoir in China [89] consisted of two sessions within one month, revealing in situ values ranging from 2.13 to 142 NTU.

Waterbodies with turbidity levels in a very wide range from 0.1 to 1000 NTU exhibit a strong correlation with satellite imagery. Research conducted over 10 years throughout six summer sessions in New Zealand [84] found a significant association ($R^2 = 0.924$) between Landsat 7 ETM+ images and in situ values from 34 shallow lakes. In situ values ranged from 75 to 275 NTU. There is a strong correlation between turbidity and PlanetScope data ($R^2 = 0.79$) in a 3-year study of 13 reservoirs in Oklahoma, USA [80], where in situ turbidity levels ranged from 0 to 966 NTU.

The literature summarized in Table S4 provides information on the most commonly used sensors for assessing turbidity from satellite and airborne multispectral and hyperspectral imagery. The Landsat-5 TM sensor is frequently used and has an average $R^2$ value of 0.82 based on five studies. However, other satellite sensors such as Landsat-7 ETM+, Terra ASTER, and PROBA-CHRIS have achieved better results with $R^2$ values greater than 0.9 for assessing turbidity. Deep waterbodies (>15 m) have the highest $R^2$ values of 0.88 for retrieving turbidity from satellite and airborne imagery. Small-sized waterbodies (<1 km$^2$) achieve the best results with an $R^2$ value of 0.98. Studies lasting between two and six months have shown the best results, with an $R^2$ value greater than 0.87. The most effective methods for extracting turbidity from satellite and airborne images are empirical methods and NN ($R^2 > 0.85$).

According to the literature in Table S4, the most effective sensor for measuring turbidity in small (<1 km$^2$) and large (10–100 km$^2$) waterbodies is Landsat-5 TM. For very large waterbodies (>100 km$^2$), the recommended sensors are multispectral Landsat-5 TM, Landsat-7 ETM+, and hyperspectral PROBA-CHRIS. Multispectral Landsat-5 TM is also the most effective for measuring turbidity in very shallow (<3 m) and shallow (3–15 m) waterbodies, while the most effective sensor for retrieving turbidity in deep waterbodies (>15 m) is hyperspectral PROBA-CHRIS.

### 4.3. Transparency (Secchi Disk Depth (SDD))

Water transparency, which is a measure of the clarity of lake water, is an important indicator of water quality and the health of aquatic ecosystems [90]. It is commonly assessed using the Secchi disk [91], a white and black disk that is lowered into the water until it is no longer visible. However, this method is labour-intensive and limited in its ability to capture spatial variations in water clarity. Remote sensing methods, which use satellite data to estimate water transparency, offer a more efficient and comprehensive approach by approximating Secchi disk depth in water with an inverse variation of the diffuse attenuation coefficient ($K_d$). The diffuse attenuation coefficient of downwelling irradiance, which is frequently measured at 490 nm, shows the exponential drop in irradiance with increasing water depth [46].

These methods use semi-analytical, empirical [51,72,84,92], and ML models to correlate water reflectance with transparency in lakes [93]. DL algorithms, such as NNs, have also been used to improve the accuracy of water quality retrieval models [72]. Water clarity is often used as a proxy for the trophic state of a lake [94], indicating nutrient availability and chlorophyll concentrations. Turbidity and TSM levels in the water are inversely correlated with water clarity. Various spectral bands and ratios are used in remote sensing to measure water clarity, with wavelengths in the red spectrum being particularly effective [84]. Table S5 summarizes these findings. Landsat-5 TM and Envisat MERIS satellite systems have been found to be effective for evaluating water clarity due to their comparatively low cost, temporal coverage, spatial resolution, and data availability.

There is a significant correlation between satellite imagery and waterbodies, with SDD values in a narrow range of 0.1 and 2 m. An analysis of Landsat-5 TM data showed a moderate correlation ($R^2 = 0.588$) between SDD in situ values ranging from 0.16 to 0.33 m during a 1-month period on a lake in Tennessee, USA [79]. For a Chinese lake [72], Terra MODIS images revealed a moderate correlation ($R^2 = 0.628$) with in-situ SDD measurements ranging from 0.25 to 1.2 m over the course of four months. Using in-situ measurements ranging from 0.23 to 0.39 m over the course of a month, SDD and Envisat MERIS imagery of a South African lake [95] show a strong correlation with a $R^2$ value of 0.801. A month-long investigation on an Italian lake [75] found a strong correlation between SDD and Landsat-5 TM ($R^2 = 0.82$) with in situ values ranging from 0.25 to 1 m. A 1-month study on three lakes in Brazil's Lower Amazon Floodplain [92] using SDD and PlanetScope imagery revealed a strong correlation ($R^2 = 0.816$) with in situ values ranging from 0.6 to 1.94 m.

Waterbodies exhibiting a medium range of SDD values between 0.1 and 3.75 m demonstrate a significant correlation with particular satellite images. A study conducted on 34 shallow lakes in New Zealand [84] identified a moderate correlation ($R^2 = 0.67$) between Landsat-7 ETM+ data and SDD in situ values varying from 0.05 to 3.04 m over the course of six summer sessions spanning ten years. SDD and identical satellite imagery have a strong correlation ($R^2 = 0.8$), as demonstrated by a study conducted for three months in the summer at a lake on the Canada–United States border [96] using in situ values ranging from 0.1 to 3 m. A strong correlation ($R^2 = 0.82$ and $0.929$) was observed between SDD and Landsat-5 TM imagery for one lake in Italy [75] and one lake in Thailand [97], respectively. The 1-month study conducted in Italy [75] has measured in situ values from 3 to 3.75 m. For a study conducted in Thailand [97] over the course of three spring sessions in two months, the values ranged from 0.2 to 2.5 m.

Waterbodies exhibiting a wide range of SDD values between 0.1 and 15 m demonstrate a moderate-to-strong correlation with particular satellite images. A study conducted on a lake in Spain [77] identified a moderate correlation ($R^2 = 0.63$) between Landsat-5 TM data and SDD in situ values spanning a duration of six years, with values varying from 1.33 to 7.53 m. The correlation between SDD and Terra MODIS imagery is moderate ($R^2 = 0.52$), as demonstrated by a 1-month study involving 15 lakes in Minnesota, USA [81], with in situ values ranging from 0.2 to 6.1 m. In three years, Sentinel-2 MSI imagery revealed a strong correlation ($R^2 = 0.8$) between in situ SDD values ranging from 0.08 to 4 m for 13 reservoirs in Oklahoma, USA [80]. Multiple studies [76,78,98] have found a significant correlation ($R^2 > 0.82$) between SDD and Landsat-5 TM/Landsat-7 ETM+, with SDD in situ values varying from 0.02 to 6.8 m and study durations spanning from one month to two years. There is a strong correlation ($R^2 = 0.95$) between SDD and PROBA-CHRIS, with in situ values ranging from 0.1 to 6 m, according to a 1-month study of ten lakes in Poland [99]. A strong correlation ($R^2 = 0.989$) was observed between SDD and Ikonos OSA imagery and in situ values ranging from 0.8 to 6.5 m in a Turkish estuary during a 1-month study [100]. Two Finnish studies [36,101] demonstrate a strong correlation ($R^2 > 0.86$) between SDD and AISA imagery within the in situ range of 0.3 to 7 m.

The literature summarized in Table S6 provides information on the most commonly used sensors for assessing SDD from satellite and airborne imagery. The Landsat-5 TM sensor is frequently used and has an average $R^2$ value of 0.8 based on seven studies. However, other satellite sensors such as Ikonos OSA, PROBA-CHRIS, and airborne AISA have achieved better results with $R^2$ values greater than 0.87 for assessing SDD. Deep waterbodies (>15 m) have the highest $R^2$ values of 0.88 for retrieving SDD from satellite and airborne imagery. Large waterbodies (10–100 km$^2$) achieve the best results with an $R^2$ value of 0.92. Studies lasting between two and three months have shown the best results, with an $R^2$ value of 0.87. The most effective method for extracting SDD from satellite and airborne images is through empirical methods and multiple regression ($R^2 > 0.82$).

The literature in Table S6 suggests that the most effective sensor for measuring SDD in small (<1 km$^2$) and very large (>100 km$^2$) waterbodies is Landsat-5 TM. For large waterbodies (10–100 km$^2$), both multispectral Landsat-5 TM and Ikonos OSA are recommended.

Hyperspectral Envisat MERIS is the recommended sensor for medium-sized waterbodies (1–10 km$^2$) and very shallow waterbodies (<3 m). Landsat-5 TM is most effective for assessing SDD in shallow waterbodies (3–15 m), while multispectral Landsat-5 and Ikonos OSA are most effective for assessing SDD in deep waterbodies (>15 m).

*4.4. Water Temperature (WT)*

WT is an important indicator of ecosystem health and water quality [14]. Accurate surface WT measurements are crucial for weather and climate research, and remote sensing can provide these measurements. However, measurements can be affected by factors like emissivity and atmospheric absorptions [102]. Infrared radiometers can provide surface WT measurements with a precision of around 0.5 °C, but optical remote sensing methods should be used to identify and mask clouds and fog. Passive microwave approaches can be used in cloudy locations with an accuracy limit of roughly 1.5–2 °C [103]. While passive microwave radiometers have lower accuracy and resolution compared to infrared radiometers, they are not affected by air and cloud influences [9]. Estimating primary production and phytoplankton growth rates can be performed using remote sensing and in situ measurements of WT [14]. WT also affects DO concentrations and the distribution of contaminants in the water. Remote sensing, combined with in situ measurements, can provide accurate data on temperature zones at a reasonable cost. Various studies have explored the challenges and benefits of using empirical methods [83,104] and numerical weather prediction models to estimate WT in different types of lakes. Following a conducted review of the literature, Table S7 summarizes how combinations of bands may be used to measure WT. The most commonly utilized instruments mounted on satellites used for remote sensing retrieval of WT are Landsat-8 TIRS and Terra MODIS.

A correlation has been observed between specific satellite imagery and waterbodies exhibiting WT concentrations that have narrow-range variability. An R-squared value of 0.535 indicates a moderate correlation between WT and Terra ASTER for a lake in Egypt [83], where WT levels fluctuate between 29.7 and 31.2 °C over the course of two months.

Waterbodies characterized by wide range in WT levels demonstrate a significant correlation with satellite imagery. Over a 10-month period, Landsat-7 ETM+ and Landsat-5 TM satellite imagery established a strong correlation ($R^2 = 0.921$) with WT in multiple lakes located in northern Germany [105]. The in situ temperatures measured during this period varied between 2.5 and 21.5 °C. Based on 120 sessions over six years and in situ values spanning from 1 to 29 °C, a study on four lakes in Switzerland–France, Hungary, Sweden, and Finland [106] reveals a strong correlation ($R^2 = 0.792$) between WT and NOAA-9, -11, -12, -14, -16, -17, and-19 AVHRR. A strong correlation ($R^2 = 0.92$ and 0.9928, respectively) is observed between Terra MODIS and WT in two lakes in Sweden [107] and one lake in Iran [104]. The research in Sweden [107] spanned two years from April to October and utilized in situ temperatures varying from 1 to 22 °C. The Iranian study [104] spanned four years and utilized in situ temperatures ranging from 3.5 to 32 °C.

The literature summarized in Table S8 provides information on the most commonly used sensors for assessing WT from satellite imagery. The Terra MODIS sensor is frequently used and has an average $R^2$ value of 0.96 based on two studies, making it the most effective sensor for retrieving WT. Medium-sized waterbodies (1–10 km$^2$) have the highest $R^2$ value of 0.92 for retrieving WT from satellite imagery. Deep waterbodies (>15 m) achieve the best results with an $R^2$ value of 0.96. Studies lasting more than six months have shown the best results, with an $R^2$ value greater than 0.89. The most effective method for extracting WT from satellite images is through the empirical method, which has an $R^2$ value of 0.84.

The literature in Table S8 suggests that different sensors are recommended for assessing WT in different sizes and depths of waterbodies. For very large waterbodies (>100 km$^2$), Terra MODIS is considered the most effective sensor. For medium-sized waterbodies (1–10 km$^2$), multispectral Landsat-5 TM and Landsat-7 ETM+ are the recommended sensors. Landsat-7 ETM+ and Terra MODIS are the most effective for assessing WT in shallow (3–15 m) and deep waterbodies (>15 m).

### 4.5. Salinity

Salinity is an important parameter for brackish lakes but not relevant for freshwater lakes. It affects water density and currents, as well as the exchange of gases between air and water. Satellite measurements can be affected by a layer of fresh surface water on top of salty water. Salinity in inland waterbodies varies due to factors like precipitation, evaporation, river runoff, and interactions with oceans [14]. Monthly salinity maps can help determine variations in freshwater input and outflow. Table S9 lists the band combinations needed to accurately measure salinity using optical sensors on satellites like Landsat-5 TM, Landsat-8 OLI, and Sentinel-2 MSI. Table S9 presents a concise overview of the studies conducted to assess salinity in shallow lakes through empirical [83], ML [108], and NN methods [109]. Indirect methods, such as brightness temperature, CDOM, and temperature profiles, are used to estimate salinity. Due to the lack of a direct colour signal from salinity, the colour signal can instead be estimated using relationships between salinity, WT, and brightness temperature [110] and between salinity and CDOM [111].

An R-squared value of 0.657 indicates a moderate correlation between Sentinel-2 MSI imagery and salinity for a hypersaline lake in Iran [108], where salinity values fluctuate between 30.7 and 36.1 over the course of three months (April, June, and July 2021). In April and June of 2019, a strong correlation ($R^2 = 0.94$) was identified between salinity measurements obtained from Sentinel-2 MSI in the same lake in Iran [109]. In situ values for this correlation varied from 6.5 to 32.

The literature on remote sensing-based salinity assessment in lakes is limited, with only two studies conducted on Urmia Lake, a very large and shallow waterbody. Both studies used the Sentinel-2 MSI sensor and found promising results, as highlighted in Table S10. The first study, which lasted for three months, used the ML method and achieved an $R^2$ value of 0.66. The second study, which lasted for two months, used the NN method and achieved a high $R^2$ value of 0.94.

### 4.6. Electrical Conductivity (EC)

The EC of water is a measure of its ability to conduct electricity and is influenced by the concentration of ions or salt in the water [112]. The standard unit of measurement for EC is microSiemens per centimetre (μS/cm). Higher salinity levels in water lead to a decrease in oxygen absorption. Changes in EC that occur rapidly can indicate water contamination. Anions like chloride, phosphate, and nitrate can increase EC when added to sewage discharge or agricultural runoff [112]. The combination of chemical and biological processes can cause changes in EC, and diurnal variations in EC have been observed during low-flow cycles [113]. Conductivity probes are used to measure EC in the laboratory or field, and some devices can also measure salinity. Correlations between EC and spectral measurements are challenging due to complex interactions with optically active water quality elements [32]. Table S11 lists selected remotely taken measurements of EC using various sensors and spectral bands and band combinations. Landsat-8 OLI is commonly used for retrieving EC, either as a single band or in combination with other bands.

Waterbodies with EC in a narrow range from 0.01 to 4 mS/cm exhibit a moderate correlation with specific satellite images. An R-squared value of 0.699 indicates a moderate correlation between EC and Landsat-8 OLI imagery for a reservoir in Columbia [88] (EC values range from 0.54 to 1.82 mS/cm over one month) and an R-squared value of 0.615 for the same satellite imagery in a lake in Kashmir, India [86] (EC levels range from 0.01 to 0.3 mS/cm over one month).

Waterbodies with EC in the medium range of 40 to 60 mS/cm exhibit a strong correlation with specific satellite imagery. A strong correlation was identified between EC and Landsat-8 OLI in a lake in Egypt [114] over the course of one month, with in situ values varying from 42.86 to 52.55 mS/cm and coefficient of correlation $R^2 = 0.87$.

The literature summarized in Table S12 provides information on the most commonly used sensors for assessing EC from satellite imagery. The Landsat-8 OLI sensor is frequently used and has been found to be the most effective sensor for retrieving EC, with an average

$R^2$ value of 0.73 based on three 1-month studies [86,88,114]. These studies were conducted on very large (>100 km$^2$) and shallow waterbodies (3–15 m). The most effective method for extracting EC is regression, which has an $R^2$ value of 0.87 [114].

## 5. Sensors for Assessing Water Quality Parameters

The complementary use of traditional in situ monitoring and remote sensing data/products maximizes strengths and minimizes existing weaknesses in lake monitoring. Satellite and airborne (aircraft and unmanned aerial vehicle (UAV)) remote sensing methods are important for evaluating the quality of inland waterbodies [14]. To monitor water quality over time, it is necessary to calibrate and validate satellite and airborne data using in situ measurements. Different types of sensors on UAVs, aircrafts, and satellites (Table 2) can analyse waterbody radiation at different wavelengths and scales. Multispectral and high-resolution remote sensing devices record reflected or emitted radiation in a few spectral bands that cover a considerable section of the electromagnetic spectrum for inland water quality monitoring [115–118]. Hyperspectral sensors measure continuously across the electromagnetic spectrum in up to 200 narrow spectral bands [119]. Due to their high spatial and spectral resolutions and simultaneous collection of narrower and contiguous bands, hyperspectral sensors can measure and monitor many water quality parameters in lakes [36,101,120]. Spaceborne sensors with visible, infrared, and microwave wavelengths can also monitor water quality. UAVs integrated with various sensors are practical and efficient for water management and can accurately recover water quality parameters due to its higher spatial and spectral resolution for smaller waterbodies. Data fusion from multiple satellite sensors can provide higher spatial, temporal, and spectral resolution for water quality monitoring [121]. Atmospheric correction and addressing adjacency effects are important for post-processing remote sensing data. Atmospheric correction reduces atmospheric radiation error and improves the evaluation of water quality parameters [8]. UAVs with high-resolution sensors can measure water quality without atmospheric impacts [122]. Factors such as white caps, sun glare, wave motion, and vegetation density can affect remote sensing imagery processing [46]. Reflection and refraction can be limited by collecting data during calm conditions and using a nadir sensor setup [123]. Bathymetric data enhances water column correction by providing information on the depth of waterbodies.

The spatial, temporal, and spectral resolution limitations of numerous contemporary satellite and airborne sensors can restrict the use of remotely sensed data for evaluating water quality. Satellite sensors are preferable for large and very large waterbodies, while airborne and UAV sensors are effective in collecting frequent and wide-ranging data for small and medium-sized waterbodies [124]. Nonsatellite remote sensing data is less affected by atmospheric conditions. The cost of hyperspectral or airborne data is one of the primary limitations of using these remote sensing methods for assessing water quality. UAV remote sensing data collection is challenging due to limitations in flight duration, weather conditions, and the data requirements for creating high-quality orthomosaic maps [124]. Remote sensing technologies, such as satellite and airborne sensors, are useful for collecting historical lake ecological indicator data in unstudied lakes without monitoring networks or data. The project budget, spatial and spectral resolution, and geographic coverage area determine the remote sensing platform. Table 2 includes regularly used satellite and airborne sensors in aquatic environments.

**Table 2.** Overview of satellite and airborne sensors commonly used in aquatic environments.

| Satellite Sensor | Full Name of the Sensor | Platform | Sensor Type | Agency | Operational Years | Reference |
|---|---|---|---|---|---|---|
| AISA | Airborne Imaging Spectrometer for Applications | Airborne | Hyperspectral | Specim | - | [125] |

**Table 2.** *Cont.*

| Satellite Sensor | Full Name of the Sensor | Platform | Sensor Type | Agency | Operational Years | Reference |
|---|---|---|---|---|---|---|
| CASI | Compact Airborne Spectrographic Imager | Airborne | Hyperspectral | Itres Research | - | [126] |
| Daedalus ATM | Airborne Thematic Mapper | Airborne | Multispectral | Daedalus Enterprises | - | [127] |
| HyMap | - | Airborne | Hyperspectral | NASA | - | [128] |
| HyperOCR | Ocean Colour Radiometer | Airborne | Hyperspectral | Sea-Bird | - | [129] |
| MIVIS | Multispectral Infrared and Visible Imaging Spectrometer | Airborne | Hyperspectral | Italian National Research Council | - | [130] |
| Envisat MERIS | Medium Resolution Imaging Spectrometer | Satellite | Hyperspectral | ESA | 2002–2012 | [131] |
| EO-1 Hyperion | - | Satellite | Hyperspectral | NASA | 2000–2017 | [132] |
| Ikonos OSA | Optical Sensor Assembly | Satellite | Multispectral | GeoEye | 1999–2015 | [133] |
| ISS HICO | Hyperspectral Imager for the Coastal Ocean | Satellite | Hyperspectral | NASA | 2009–2014 | |
| Landsat-5 MSS | Multi-Spectral Scanner | Satellite | Multispectral | NASA | 1972–2011 | [134] |
| Landsat-5 TM | Thematic Mapper | Satellite | Multispectral | NASA | 1982–2011 | [134] |
| Landsat-7 ETM+ | Enhanced Thematic Mapper Plus | Satellite | Multispectral | NASA | 1999–present | [135] |
| Landsat-8 OLI | Operational Land Imager | Satellite | Multispectral | NASA | 2013–present | [136] |
| Landsat-8 TIRS | Thermal Infra-Red Sensor | Satellite | Multispectral | NASA | 2013–present | [136] |
| NOAA AVHRR | Advanced Very High-Resolution Radiometer | Satellite | Radiometer | NOAA | 1998–present | [137] |
| PlanetScope | - | Satellite | Multispectral | Planet | 2014–present | [138] |
| PROBA-CHRIS | Compact High Resolution Imaging Spectrometer | Satellite | Hyperspectral | UKSA | 2002–present | [139] |
| Sentinel-2 MSI | Multispectral Instrument | Satellite | Multispectral | ESA | 2015–present | [140] |
| Sentinel-3 OLCI | Ocean and Land Colour Instrument | Satellite | Multispectral | ESA | 2016–present | [141] |
| Terra ASTER | Advanced Spaceborne Thermal Emission and Reflection Radiometer | Satellite | Multispectral | NASA | 2000–present | [142] |
| Terra MODIS | Moderate Resolution Imaging Spectroradiometer | Satellite | Multispectral | NASA | 2000–present | [143] |
| WorldView-2 | - | Satellite | Multispectral | DigitalGlobe | 2010–present | [144] |

## 6. Discussion and Recommendations

This paper discusses the use of remote sensing methods for monitoring water quality in lakes. The paper focuses on three main areas: bibliometric analysis of published literature, methods for retrieving water quality using remote sensing, and exploring optically active water quality parameters that may be assessed using remote sensing.

The literature reviewed in this study provides information on the most commonly used sensors and methods for assessing various water quality parameters from satellite and airborne imagery. The Landsat-5 TM sensor is frequently utilized and has a consistently significant value of $R^2$ for all parameters, regardless of the waterbody size and depth. Different sensors and methods have achieved the best results for different parameters, with some sensors consistently performing well across multiple parameters. For small waterbodies, the most effective satellite sensor for chl-$\alpha$, turbidity, and SDD retrieval is Landsat-5 TM. For chl-$\alpha$ retrieval from small and medium-sized waterbodies, the most effective airborne sensors are CASI and HyMap, whereas the most effective satellite sensor for recovering chl-$\alpha$ and SDD for medium waterbodies is MERIS. For large waterbodies, the most effective satellite sensor for assessing chl-$\alpha$ and SDD is the Ikonos OSA, while the Landsat-5 TM is the most effective sensor for retrieving turbidity and SDD. The most effective satellite sensors for chl-$\alpha$ retrieval from very large waterbodies are hyperspectral MERIS and multispectral Sentinel-3 OLCI. The most effective sensors for turbidity and SDD retrieval from very large waterbodies include multispectral Landsat-5 TM and Landsat-7 ETM+, and hyperspectral PROBA-CHRIS. For very large waterbodies, the most effective sensor for retrieving WT is MODIS, for salinity is Sentinel-2 MSI, and for EC is Landsat-8 OLI.

The retrieval of water quality parameters using remote sensing can be achieved through various methods. One commonly used method is the analytical method, also known as the physical method, which is characterized by its theoretical analyses of spectral data. Statistical analyses are commonly used in empirical and semi-empirical methods, which are preferred due to their complexity. On the other hand, ML methods, which are empirical in nature, are known for their computational complexity and ability to manage nonlinear relationships. NNs are essential components of ML and have gained significant importance in solving different tasks in supervised ML [145]. The most successful methods for retrieving chl-$\alpha$ and turbidity from satellite and airborne imagery are the empirical and NN methods, with a high coefficient of correlation ($R^2$) value of 0.98. The most effective method for extracting salinity from satellite and airborne images is the NN method with a high correlation coefficient ($R^2 = 0.94$). Regression is the most effective algorithm for retrieving SDD and EC, while the empirical method is the most suitable for retrieving WT.

Satellite data is acknowledged as a useful tool for monitoring water quality parameters in lakes. Unlike in situ measurements, satellite imagery gather water quality data simultaneously using a grid-based method. The concept of using remote sensing technology for water quality monitoring is based on the different spectral characteristics of pure water compared to contaminated or saturated water. The properties of individual water quality parameters are analysed in relation to their interaction with the spectrum to identify bands. These bands are combined to obtain the parameter's value and its distribution over the lake [14].

However, there are limitations to remote sensing methods, including spatial, temporal, and spectral resolutions of satellite systems, the optical complexity of inland waters, atmospheric and cloud interference, the need for proper calibration and validation with in situ measurements [14], errors in creating standard satellite products like atmospheric correction [146], and the cost of commercial satellite imagery or deploying aircrafts or UAVs for study purposes. Various satellite systems, such as Landsat, Sentinel-2, and Terra, are used in the literature to estimate water quality metrics, but for smaller lakes, the selection of available satellite sensors is limited. Satellites with spatial resolution like Terra MODIS (260 m, 500 m, and 1000 m), Envisat MERIS (260 m $\times$ 300 m), or OrbView-2 SeaWiFS (260 m, 500 m, and 1000 m) are not recommended due to their tendency to overgeneralize the state of the parameters. This especially applies to parts of smaller lakes with stronger external influences (the influence of ballast water or nutrients from agricultural land). For the purpose of determining amplitude values on an annual level, bands with a higher spatial resolution can be used; however, if smaller changes in parameters are to be observed in shorter periods of time (day, week, month), it is necessary to provide bands with a better

spatial resolution. UAVs with integrated sensors are effective for water management and can accurately measure water quality parameters in smaller waterbodies. Their higher spatial and spectral resolution makes them practical and efficient for this purpose. The selection of satellite and aerial images should depend on the dynamics of parameter changes within the lake and the lake's size, rather than just the availability of images.

To conduct effective remote research, it is important to familiarize oneself with the lake and its environment, understand the seasonal cycle of submerged macrophytes and phytoplankton, identify external sources of water flow into the lake, consider meteorological conditions during specific times of the year (including water temperature, water levels, dry spells, and high-water periods following heavy rainfall prior to measurement), and understand agricultural practices around the lake. The correlation between satellite and airborne imagery and in situ values varies depending on the size and depth of the waterbodies. The research suggests that small waterbodies have the highest correlation for retrieving turbidity, medium waterbodies for chl-$\alpha$ and WT, large waterbodies for SDD, and very large waterbodies for salinity and EC. Deep waterbodies have the highest correlation for retrieving WT, turbidity, and SDD, while very shallow waterbodies have the highest correlation for assessing chl-$\alpha$, and shallow waterbodies have the highest correlation for retrieving salinity and EC. Extensive temporal statistical analysis of in situ data (specific water quality parameters) along with meteorological and hydrological data (water level, lake depth, and dry and rainy periods) is recommended to identify correlations and mitigate negative effects. Modelling based on specific conditions can provide insight into the movement of parameter concentrations on the lake's surface. The development of ML and NN methods aligns well with this scenario, as it leverages the progress in computer technology and storage capacity to enhance productivity and retain data for future studies. In this way, a system is established that acquires and applies all acquired "knowledge" (representing stored results) for subsequent analyses.

The range of a specific parameter measured in situ inside the lake throughout the study duration and the length of the study (year, season, month, week, and day) are crucial pieces of information for estimating water quality parameters based on remote sensing. The time period with the highest correlation between satellite and airborne imagery and in situ values varies for different water quality parameters. The highest correlation between satellite and airborne imagery and in situ values for retrieving turbidity, SDD, and salinity is found within a time frame of 2–3 months. For EC, the highest correlation is achieved in studies lasting one month, while for chl-$\alpha$ and WT, it takes studies lasting more than six months up to several years to obtain the highest correlation. When water quality measurements are determined annually (over twelve months), all influences on the lake (both internal and external) are averaged. The parameter values are observed throughout the year, but variations due to varied weather conditions (seasons and meteorological data) and other external parameters are not included. The purpose of a forecasting model is to predict extreme events that significantly impact the lake, such as heavy rainfall during intense agricultural activity or the intrusion of seawater during an extremely dry period, in order to minimize their impact on the lake's water quality. Following a thorough examination of how external influences impact internal parameters and the dispersion of water quality parameters in the lake, it is essential to choose suitable satellite or airborne data to achieve optimal outcomes. This pertains to the spatial resolution of imagery to represent the frequency of changes in water quality parameter values, the spectral bands of sensors for calculating optically active water quality parameters, and the temporal resolution of the system. Ensuring that in situ measurements are taken on the day of the satellite's overpassing (or in a window of a few days prior to or after the overpass) allows for conducting correlation analysis between the observed parameter values from in situ measurements and calculated parameter values from satellite data.

The synergy of computer resources, remote sensing methods and data, GIS multicriteria analysis, decision support systems, and ML methods enable a high-quality assessment of water quality parameters. The authors' knowledge collected in this review and previous

general and specific knowledge about the considered problem and the field of remote sensing resulted in the creation of a framework for monitoring water quality parameters in the lake, supported by remote sensing methods (Figure 4).

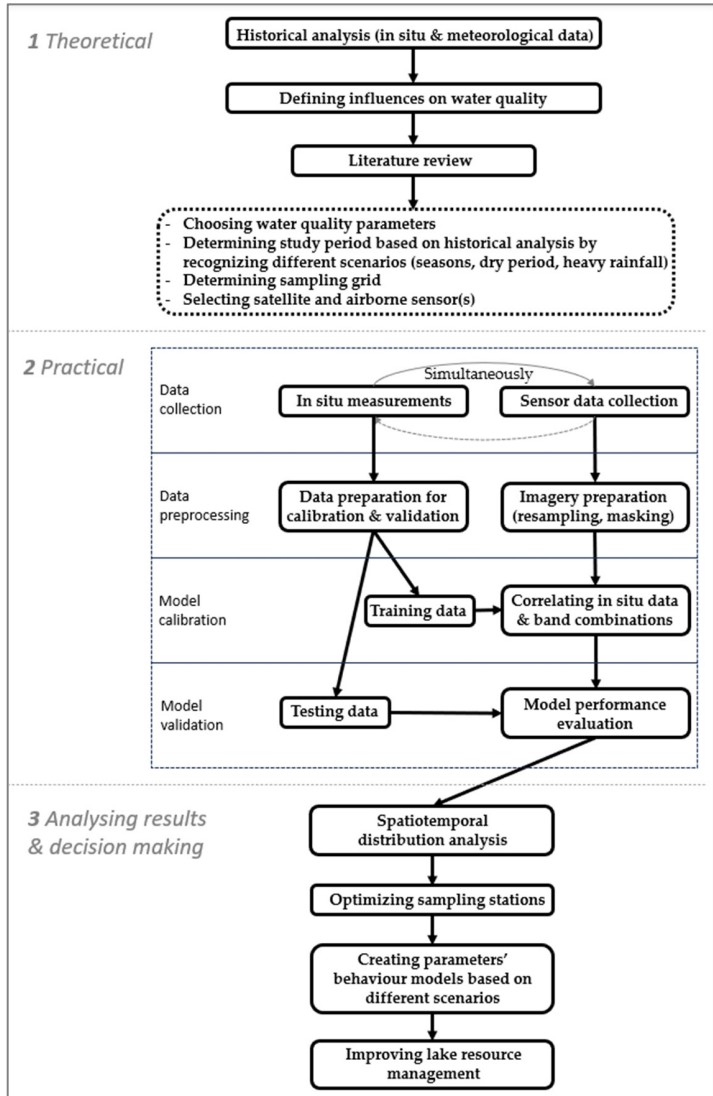

**Figure 4.** Workflow proposal for monitoring and assessment of water quality parameters in lakes using remote sensing methods.

The proposed working framework (Figure 4) includes three main components: (1) conducting a literature review to gain theoretical knowledge, (2) practical work involving data collection and analysis, and (3) analysing the results for decision-making. The theoretical part of the workflow includes the following:

(1) Conducting a thorough time analysis of the waterbody and its surroundings (for a period of 10 to 15 years, utilizing meteorological and in situ data) under various conditions (dry period, rainfall period, etc.).

(2) Determining and evaluating all internal and external factors that may influence water quality.

(3) Referring to relevant scientific research.

(4) Determining which water quality parameters will be included in the study.

(5) Determining the study period based on historical analysis by recognizing various environmental scenarios such as seasons, dry periods, and heavy rainfall. If a lake is susceptible to ice or snow, the study period should focus on the coldest months when

the lake is affected by these conditions. Similarly, if a brackish lake experiences dry periods leading to increased salinity due to evaporation and low water influx, the study period should include this phenomenon.

(6) Analysing temporal statistical data and water quality parameter distribution throughout the waterbody, taking into account the hydrological model of the lake's bathymetry and important tributaries, to determine the sampling grid and the quantity and locations of in situ measurement locations.

(7) Choosing the appropriate satellite or airborne sensor(s) for data collection based on spectral characteristics, number of bands, spatial resolution, time resolution (if the system is a satellite), lake size and depth, and chosen water quality parameters.

The operational features of the workflow can be determined by considering certain factors, as follows:

(1) In situ measurement collection is based on a defined study period and sampling grid; data collection should occur on the day of the chosen sensor's overpassing or in a small window frame ($\pm 4$ days) around that day.

(2) In situ data should be analysed by removing outliers and normalizing the data. This is important in order to make the measured values for different parameters comparable, even if they are measured in different units (e.g., chl-$\alpha$ is usually measured in $\mu g/L$, while WT is measured in $°C$).

(3) A total of 30% of in situ measurements shall be utilized for validation purposes, while the remaining 70% shall be utilized to calibrate the calculated values of water quality parameters via the spectral band combinations of the chosen sensor(s).

(4) To ensure accurate remote sensing data, it is important to collect the data using a suitable satellite, aircraft, or UAV platform.

(5) All imagery should be resampled to the same spatial resolution and undergo necessary corrections such as geometric, radiometric, and atmospheric correction. Any obstructions like clouds, haze, or other obstacles covering water pixels on an image should be masked out. Satellite measurements use a grid-based method to gather water quality data simultaneously, so the reflectance values of different water sampling locations are extracted to analyse the spectral characteristics.

(6) Remote sensing technology is used to monitor water quality by analysing the interaction of water quality parameters with the spectrum and identifying specific bands. These bands are combined as single bands, band ratios, and band combinations to obtain the parameter's value and its distribution over the lake.

(7) A correlation analysis is performed between measured in situ data and band combinations from selected sensors. This analysis is conducted on a training dataset in order to determine the best method (e.g., analytical, empirical, or ML), which is the one with the highest correlation coefficient.

(8) The developed models are validated using 30% of in situ measurements as a testing dataset.

The final part of the workflow involves analysing the results for decision-making, as follows:

(1) Conducting spatiotemporal distribution analysis and generating accurate spatial distribution maps based on validation results.

(2) Optimizing sampling locations by using spatiotemporal analysis and GIS multicriteria analysis. This involves considering various influencing factors such as key water quality parameters, meteorological data, and environmental influences (e.g., distance to the tributaries and land cover-land use data).

(3) Creating a database of outputs for different lake environment scenarios, which can be used by ML methods to simulate and forecast lake behaviour in similar scenarios.

(4) Recommendations to the authorities in charge of managing the lake on how to improve lake resource management based on data collection, analysis, and modelling. Developed models can be used by local authorities to obtain surface water quality pa-

rameters of the lake during periods with similar weather conditions as the ones used for model generation. This approach offers reduced cost and time while maintaining reasonable accuracy.

## 7. Conclusions

Conducting effective remote research on lakes requires a comprehensive understanding of the lake and its environment, including factors such as submerged macrophytes, phytoplankton, hydrology, bathymetry, meteorological conditions, and agricultural practices. Analysing year-round fluctuations from historical water quality data and categorizing data based on common criteria can help identify significant seasonal variations. The use of satellite and airborne imagery, along with in situ data and statistical analysis, can help identify correlations and mitigate negative effects. Empirical methods are commonly used for monitoring and analysing water quality due to the complexity of analytical methods, but ML and NN methods show promise in simplifying and analysing large datasets with high accuracy. Modelling based on specific conditions can provide valuable insights into parameter concentrations on the lake's surface. By leveraging advancements in computer technology and storage capacity, a system can be established to acquire and apply knowledge for future analyses. Overall, remote research methods offer valuable insights into lake ecosystems and water quality.

The proposed workflow aims to provide guidance for effectively monitoring and managing the quality of lakes. It combines theoretical knowledge about the study area, practical work regrading data collection, and data analysis to provide a comprehensive approach to understanding and improving water quality. Furthermore, the framework includes conducting thorough analysis, choosing appropriate sensors, and incorporating remote sensing technology to generate accurate water quality models and spatial distribution maps. The output generated by the workflow has multiple benefits, including scientific purposes, decision-making, and resource management. The proposed framework aims to enhance global water quality monitoring by integrating different data sources and methods to understand spatiotemporal water quality trends.

The review highlights the importance of integrating remote sensing methods using in situ measurements and computer modelling to improve the understanding of water quality. Future research should focus on (1) the development of advanced technologies, such as advanced algorithms, for in-depth statistical analyses of data (meteorological and measured in situ water quality parameters); (2) selection of a sampling grid of strategic in situ measurements (according to the distribution and concentration of a particular water quality parameter) and time period, (3) integration of very high resolution spectral data (the selection of spectral bands and increased use of hyperspectral sensors for estimating water quality parameters), and (4) integration of ML and NN algorithms for effective water quality problem solving (use of modern computer technologies in modelling different scenarios of impact on lake water). These technologies can enhance the ability to detect and respond promptly to water quality issues, optimize sampling locations and time frames, estimate optically inactive parameters indirectly, and facilitate real-time monitoring and timely response to potential risks or anomalies. Additionally, ML and NN algorithms can provide valuable insights and predictive models for future water resource management, especially in dynamic and ever-changing water systems.

**Supplementary Materials:** The following supporting information can be downloaded at: https://www.mdpi.com/article/10.3390/hydrology11070092/s1. Analysis of Water Quality Parameters Using Satellite Sensors and Spectral Bands, Tables S1–S12 can be found attached in the Supplementary Materials. References [147–177] used in the Supplementary Materials are listed in the article.

**Author Contributions:** Conceptualization, A.B. and A.K.; methodology, A.B. and A.K.; validation, A.B. and A.K.; investigation, A.B.; resources, A.B. and A.K.; writing—original draft preparation, A.B.; writing—review and editing, A.B. and A.K.; visualization, A.B.; supervision, A.K.; project administration, A.B. All authors have read and agreed to the published version of the manuscript.

**Funding:** This research received no external funding.

**Data Availability Statement:** Not applicable.

**Conflicts of Interest:** The authors declare no conflicts of interest.

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
