# Peer review of "Integrating Remote Sensing Methods for Monitoring Lake Water Quality: A Comprehensive Review"

_hydrology, doi:10.3390/hydrology11070092_

Round 1
Reviewer 1 Report
Comments and Suggestions for Authors
The work entitled "Integrating Remote Sensing Methods for Monitoring Lake Wa-2 ter Quality: A Comprehensive Review".
It is about a topic that is relevant and important to be discussed, especially nowadays with the huge boom of the use of remote sensing.
Especially for lakes it is very difficult to move to water quality analysis with the use of remote sensing and it was proved by this literature review that the remote sensing methods brings low or high accuracy results.
It is missing a map showing the location of the diferent studies.
The authors for each water quality parameter (ie chlorophyl, WT, turbidity etc) present the remote sensing methods and some statistical results of these performence, but there is not any other connection with other characteristics (time period, season, dry spell, area of the lake, drainage area etc). It would be useful for each WQ parameter a table to be done and a comparison of some methods results (ie the sentinel equations) with the main characteristics of the water bodies.
Are small lakes better studies by modis? or larger lakes better studied by sentinel? or small lakes is better to find out the temparature but not the turbidity?
it should be a better connection between the different parameters. otherwise it is just a table of various equations and is not give to the reader the information that is appropriate in order to use this equations.
however guidelines should be created according the findings of this review for small-moderate-large lakes, for rainy- non rainy period and for specific land uses about the most appropriate satellite imagery to be used.
Info is give about UAV use but this should be also extended . it is limited the information and has to be connected with the satellite imagery. Which need UAV are covering?
It is completely missing a comparison for each parameter between the different methods (empirical, semi-empirical, AI/ML). It is not obvious if AI/ML has brought better information , something new.
In the discussion the Figure 4 should be better discussed and also in the discussion part is missing much references.
more discussion about the scenarios development based on remote sensing.
The authors mention in the text
"To identify significant seasonal variations, data should be categorized based on common criteria such as water temperature, dry and wet periods, vegetation phenology, and agricultural calendars. ".
I think they should work more in that parameters (dry and wet periods, vegetaion phenology, agricultural practicies, N and P emmissions) and correlae better with the satelite imagery.
Author Response
6th June 2024
Comments on reviews and responses to reviewers
Reviewer 1
Thank you for your detailed and constructive comments. Following are the comments and our responses to them.
Reviewer's comment
‘The work entitled "Integrating Remote Sensing Methods for Monitoring Lake Water Quality: A Comprehensive Review".
It is about a topic that is relevant and important to be discussed, especially nowadays with the huge boom of the use of remote sensing.
Especially for lakes it is very difficult to move to water quality analysis with the use of remote sensing and it was proved by this literature review that the remote sensing methods brings low or high accuracy results.’
- Reviewer's comment / question
‘It is missing a map showing the location of the different studies.’
- Author's comment / answer
We have added a thematic map showing the number of literature per country of origin reviewed for the paper's bibliographic analysis.
- Reviewer's comment / question
‘The authors for each water quality parameter (i.e. chlorophyl, WT, turbidity etc) present the remote sensing methods and some statistical results of these performance, but there is not any other connection with other characteristics (time period, season, dry spell, area of the lake, drainage area etc). It would be useful for each WQ parameter a table to be done and a comparison of some methods results (i.e. the sentinel equations) with the main characteristics of the water bodies ..’
- Author's comment / answer
The tables in the supplement were created according to your comment.
- Reviewer's comment / question
‘Are small lakes better studies by modis? or larger lakes better studied by sentinel? or small lakes is better to find out the temperature but not the turbidity?
It should be a better connection between the different parameters. otherwise, it is just a table of various equations and is not given to the reader the information that is appropriate in order to use these equations.
however, guidelines should be created according the findings of this review for small-moderate-large lakes, for rainy- non rainy period and for specific land uses about the most appropriate satellite imagery to be used’
- Author's comment / answer
The tables in the supplement were created according to your comments, and additional text was added to specific chapters and the Discussion section.
- Reviewer's comment / question
‘Info is given about UAV use but this should be also extended. it is limited the information and has to be connected with the satellite imagery. Which need UAV are covering?’
- Author's comment / answer
You are right, we only mentioned the possibility of using UAV, which we briefly expanded based on your comment in the updated version of the review in chapter 5. It is not extensively explored in the review due to the challenges of processing water surface images. It is difficult to obtain a 3D model on water and sand due to the lack of recognizable details and the movement of water. This poses a challenge for current 3D rendering software.
We are currently monitoring a lake of 1.5 x 10 km, and there are restrictions on flying UAVs higher than 50 m and frequent winds further limit UAV’s use (over 100000 images!). Therefore, UAVs may not be cost-effective in all cases. However, if higher altitudes are allowed and a hyperspectral scanner is used, processing such images could be beneficial despite the mentioned problems. These reasons explain why the manuscript only briefly mentions UAVs as a possibility without delving into further detail.
- Reviewer's comment / question
‘It is completely missing a comparison for each parameter between the different methods (empirical, semi-empirical, AI/ML). It is not obvious if AI/ML has brought better information, something new.?’
- Author's comment / answer
In this version of the review, comments have been added in specific chapters and the Discussion section. Additionally, the text now uses only one term – ML (we omitted term AI).
- Reviewer's comment / question
‘In the discussion the Figure 4 should be better discussed and also in the discussion part is missing much references.
More discussion about the scenario’s development based on remote sensing.’
- Author's comment / answer
The review has been expanded to include more detailed explanations in the Conclusion and recommendations section.
- Reviewer's comment / question
‘The authors mention in the text
"To identify significant seasonal variations, data should be categorized based on common criteria such as water temperature, dry and wet periods, vegetation phenology, and agricultural calendars. ".
I think they should work more in that parameters (dry and wet periods, vegetation phenology, agricultural practices, N and P emissions) and correlate better with the satellite imagery.’
- Author's comment / answer
The review has been expanded to include more detailed explanations in the Conclusion and recommendations section.
Sincerely,
Anja Batina,
Andrija Krtalić

Reviewer 2 Report
Comments and Suggestions for Authors
This manuscript gives a good summary of the application of remote sensing in lake water with abudant references.
However, it could be more readable and beneficial to readers in several aspects.
First of all, when the authors try to pool as many previous researches as possible in terms of remotely detection and monitoring of key water quality indicators, they should organize clearly the body and show their viewpoints explicitly if possible. For example, summary of remote sensing methods of chlorophyll-a is divided into several parts regarding to its magnitude. The overlap among various chlorophyll-a range cases leads to confusion of the performance of remote sensing methods in various lakes/inland waters and also in different sensors. The same is also true with other parameters in this manuscript than chlorophyll-a. The main consequence is that readers can not get useful or instructive information from the passages unless you explicitly name it.
Secondly, when talking about and comparing performance of various remote sensing methods and of various sensors, one should not focus solely on their statistics. This is because statistics significantly depends on number, temporal and spatial coverage, and also magnitude of variables involved. In other words, the authors should clearly state those information when they want to make it understandable to readers. Also, those instances without significant statistical meaning should be discarded. One suggestion is that the authors could diseminate it from various dimensions, e.g., data type (i.e., temporal, spatial and spectral resolution), water type (water area, paramter magnitude) and also data volume used for statistics, etc.
Thirdly, if section 5 tries to give a comprehensive description of sensors capable of water quality monitoring, one should argue the availability and necessity of remote sensing data for water quality monitoring. Also, advantages and disadvantages can be analyzed in terms of satellite, airborne and in-situ data.
Besides, (1) reference citation should keept in same format (e.g., the references in Line 362 and 406). (2) it is highly recommended the abbreviation or brief introduction of various remote sensor be illustrated in the manuscript. (3) Table Sx should be Ax according to supplementary documents. (4) The first sentence (Line 296-296) seems not to be seen explicitly from Table A1. (5) Those purely stating in-situ measurements are not significantly relevant with the topic (e.g., those last sentences in sections of turbidity, transparency, water temperature, etc. Should they be discarede?
Comments on the Quality of English LanguageEnglish writing is good except several minor mistakes.
(1) Line 254-255, 'physical optically active' goes weird.
(2) Sentence in Line 536-537 and 538-539 appears excess or meaningless.
Author Response
6th June 2024
Comments on reviews and responses to reviewers
Reviewer 2
Thank you for your detailed and constructive comments. Following are the comments and our responses to them.
Reviewer's comment
‘This manuscript gives a good summary of the application of remote sensing in lake water with abundant references.
However, it could be more readable and beneficial to readers in several aspects’
- Reviewer's comment / question
‘First of all, when the authors try to pool as many previous researches as possible in terms of remotely detection and monitoring of key water quality indicators, they should organize clearly the body and show their viewpoints explicitly if possible. For example, summary of remote sensing methods of chlorophyll-a is divided into several parts regarding to its magnitude. The overlap among various chlorophyll-a range cases leads to confusion of the performance of remote sensing methods in various lakes/inland waters and also in different sensors. The same is also true with other parameters in this manuscript than chlorophyll-a. The main consequence is that readers cannot get useful or instructive information from the passages unless you explicitly name it’
- Author's comment / answer
The manuscript has been expanded by adding explanations, comments, and overview tables for water quality parameters and sensors according to your comments. Additionally, we have included a thematic map showing the distribution of literature from the bibliographic analysis reviewed for this manuscript.
- Reviewer's comment / question
‘Secondly, when talking about and comparing performance of various remote sensing methods and of various sensors, one should not focus solely on their statistics. This is because statistics significantly depends on number, temporal and spatial coverage, and also magnitude of variables involved. In other words, the authors should clearly state that information when they want to make it understandable to readers. Also, those instances without significant statistical meaning should be discarded. One suggestion is that the authors could disseminate it from various dimensions, e.g., data type (i.e., temporal, spatial and spectral resolution), water type (water area, parameter magnitude) and also data volume used for statistics, etc.’
- Author's comment / answer
We thank you for the comments that we took into account when creating the tables in the supplement and adding explanatory text.
- Reviewer's comment / question
‘Thirdly, if section 5 tries to give a comprehensive description of sensors capable of water quality monitoring, one should argue the availability and necessity of remote sensing data for water quality monitoring. Also, advantages and disadvantages can be analysed in terms of satellite, airborne and in-situ data.’
- Author's comment / answer
This part of the text has also been refined taking into account your comments.
- Reviewer's comment / question
‘Besides, (1) reference citation should keep in same format (e.g., the references in Line 362 and 406). (2) it is highly recommended the abbreviation or brief introduction of various remote sensor be illustrated in the manuscript. (3) Table Sx should be Ax according to supplementary documents. (4) The first sentence (Line 296-296) seems not to be seen explicitly from Table A1 (5) Those purely stating in-situ measurements are not significantly relevant with the topic (e.g., those last sentences in sections of turbidity, transparency, water temperature, etc. Should they be discarded?
- Author's comment / answer
- (1) - Thank you for pointing out the error in the previous manuscript. In the new version of the manuscript, this has been corrected.
- (2) - In chapter 5, a table was added that listed sensors along with their full names and descriptions in the text.
- (3) - Thank you for pointing out the error in the previous manuscript. In the new version of the manuscript, this has been corrected.
- (4) - A new table has been added in the supplement that complements the existing one.
- (5) - The text has been updated with the addition of new tables in the supplement and their descriptions. These new tables in the supplement provide additional context to the information mentioned in your comment.
- Reviewer's comment / question
‘English writing is good except several minor mistakes.
(1) Line 254-255, 'physical optically active' goes weird.
(2) Sentence in Line 536-537 and 538-539 appears excess or meaningless.’
- Author's comment / answer
Thank you for pointing out the errors in the previous manuscript. In the new version of the manuscript, this has been corrected.
Sincerely,
Anja Batina,
Andrija Krtalić

Round 2
Reviewer 1 Report
Comments and Suggestions for Authors
The manuscript has substantially revised. It is a significant review of the algorithms used to estimate water quality parameters with the use of remote sensing.
The authors have added in the discussion section a methodology for the proper use of the satellite data to monitor lakes water quality. This is an important contribution but I suggest better to integrate this methodology in the text.
some minor comments
Line 343-The Landsat-5 TM sensor is frequently utilized and has an average R2 value 343 of 0.76 based on nine studies. (Which studies- give reference)
Lines 354-364 also in line 418 - remove the word "best" . Maybe "more effective" or "more applicable".
Figure 3 needs to be improved. it looks very "childish".
L 576 -The lake is a very 576 large and shallow waterbody. (improve the sentence)
Lines 605-610 add missing references
The literature summarized in Table S12 provides information on the most commonly 605 used sensors for assessing EC from satellite imagery. The Landsat-8 OLI sensor is fre-606 quently used and has an average R2 value of 0.73 (reference), making it the most effective sensor for 607 retrieving EC. Very large waterbodies (>100 km2) and shallow waterbodies (3-15 m) 608 achieve the best results with an R2 value of 0.73 (reference). Studies lasting one month have shown 609 the best results, with an R2 value greater than 0.72. The most effective method for extract-610 ing WT is the empirical method and regression, which has an R2 value greater than 0.66 *-(reference).
Table 2. should be amended with more information. for instance start year of operation of the satellite, scale, applications, country of landing . it is good the reader to understance the difference between the satellites.
Finally it is not recommended to close the text with the "The proposed working framework (Figure 4)".
I suggest to move this part above and finelize with some important conclusinos.
Author Response
23th June 2024
Comments on reviews and responses to the reviewer
Reviewer
Thank you for your detailed and constructive comments. Following are the comments and our responses to them.
- Reviewer's comment
‘The manuscript has substantially revised. It is a significant review of the algorithms used to estimate water quality parameters with the use of remote sensing.’
- Reviewer's comment / question
‘The authors have added in the discussion section a methodology for the proper use of the satellite data to monitor lakes water quality. This is an important contribution but I suggest better to integrate this methodology in the text.’
- Author's comment / answer
Thank you. We agree the proposed framework is an important contribution and we added it in the end of the document as we gathered this knowledge as a result of the extensive research we did while writing this review. We partially mention it throughout the paper (in the introduction, about modelling in the section Materials and methods, and different results for each water quality parameter in their respective subsections), but mainly focus on it in renamed section Discussion and Recommendations and briefly concluding its importance in the section Conclusion.
- Reviewer's comment / question
‘Line 343-The Landsat-5 TM sensor is frequently utilized and has an average R2 value 343 of 0.76 based on nine studies. (Which studies- give reference)’
- Author's comment / answer
We added the references according to your comment.
- Reviewer's comment / question
‘Lines 354-364 also in line 418 - remove the word "best" . Maybe "more effective" or "more applicable".’
- Author's comment / answer
We adjusted word “best” with “most effective” throughout the document when referring to sensors, as per your comment.
- Reviewer's comment / question
‘Figure 3 needs to be improved. it looks very "childish".’
- Author's comment / answer
The figure shows a schematic representation that only illustrates the known flow of the electromagnetic spectrum when creating images. We reworked the image.
- Reviewer's comment / question
‘L 576 -The lake is a very 576 large and shallow waterbody. (improve the sentence)’
- Author's comment / answer
We rephrased it.
- Reviewer's comment / question
‘Lines 605-610 add missing references
The literature summarized in Table S12 provides information on the most commonly 605 used sensors for assessing EC from satellite imagery. The Landsat-8 OLI sensor is fre-606 quently used and has an average R2 value of 0.73 (reference), making it the most effective sensor for 607 retrieving EC. Very large waterbodies (>100 km2) and shallow waterbodies (3-15 m) 608 achieve the best results with an R2 value of 0.73 (reference). Studies lasting one month have shown 609 the best results, with an R2 value greater than 0.72. The most effective method for extract-610 ing WT is the empirical method and regression, which has an R2 value greater than 0.66 *-(reference).’
- Author's comment / answer
We rephrased some text and added the references according to your comment.
- Reviewer's comment / question
‘Table 2. should be amended with more information. for instance start year of operation of the satellite, scale, applications, country of landing . it is good the reader to understance the difference between the satellites.’
- Author's comment / answer
We added three more columns, primarily the responsible agency and operational years, as well as reference so more info can be easily read about each sensor.
- Reviewer's comment / question
‘Finally it is not recommended to close the text with the "The proposed working framework (Figure 4)".
I suggest to move this part above and finelize with some important conclusinos.’
- Author's comment / answer
We transferred the proposed workflow and the corresponding text to the section which we now renamed to Discussion and Recommendations. And in the section Conclusion we concluded the gain and importance of the proposed workflow in one paragraph.
Sincerely,
Anja Batina,
Andrija Krtalić
